# Application of Attribute-Based Encryption in Military Internet of Things Environment

**DOI:** 10.3390/s24185863

**Published:** 2024-09-10

**Authors:** Łukasz Pióro, Jakub Sychowiec, Krzysztof Kanciak, Zbigniew Zieliński

**Affiliations:** Faculty of Cybernetics, Military University of Technology, gen. Sylwestra Kaliskiego 2, 00-908 Warsaw, Poland; jakub.sychowiec@wat.edu.pl (J.S.); krzysztof.kanciak@wat.edu.pl (K.K.); zbigniew.zielinski@wat.edu.pl (Z.Z.)

**Keywords:** Internet of Things, blockchain, attribute-based encryption

## Abstract

The Military Internet of Things (MIoT) has emerged as a new research area in military intelligence. The MIoT frequently has to constitute a federation-capable IoT environment when the military needs to interact with other institutions and organizations or carry out joint missions as part of a coalition such as in NATO. One of the main challenges of deploying the MIoT in such an environment is to acquire, analyze, and merge vast amounts of data from many different IoT devices and disseminate them in a secure, reliable, and context-dependent manner. This challenge is one of the main challenges in a federated environment and forms the basis for establishing trusting relationships and secure communication between IoT devices belonging to different partners. In this work, we focus on the problem of fulfillment of the data-centric security paradigm, i.e., ensuring the secure management of data along the path from its origin to the recipients and implementing fine-grained access control mechanisms. This problem can be solved using innovative solutions such as applying attribute-based encryption (ABE). In this work, we present a comprehensive solution for secure data dissemination in a federated MIoT environment, enabling the use of distributed registry technology (Hyperledger Fabric), a message broker (Apache Kafka), and data processing microservices implemented using the Kafka Streams API library. We designed and implemented ABE cryptography data access control methods using a combination of pairings-based elliptic curve cryptography and lightweight cryptography and confirmed their suitability for the federations of military networks. Experimental studies indicate that the proposed cryptographic scheme is viable for the number of attributes typically assumed to be used in battlefield networks, offering a good trade-off between security and performance for modern cryptographic applications.

## 1. Introduction

In the era of rapid technological advancement, the intersection of military operations and Internet of Things (IoT) technologies has established a new research field, the Military Internet of Things (MIoT). Connected devices, ranging from unmanned vehicles to smart surveillance systems, are now integral to military operations. This continuous technological development is reshaping the landscape of modern conflicts, presenting new opportunities and challenges for armed forces worldwide. Examples of military applications of IoT include smart logistics (transportation), smart bases, soldiers’ health care, or Internet of Battle Things (IoBT) [1]. One of the main problems of MIoT deployment is acquiring, analyzing, and merging huge amounts of data from many different IoT devices and disseminating them in a secure, reliable, and context-dependent manner. The problem is compounded in these applications when the military has to interact with other institutions and organizations to form a federation, the purpose of which is to allow different cooperating parties to share some of the resources belonging to its various participants and to exchange acquired information from different sources. As military systems become increasingly interconnected, ensuring the confidentiality, integrity, and accessibility of sensitive data is paramount.

An increasing number of MIoT applications are expected to require the ability to operate in a federated environment. Examples include the formation of federations of NATO and non-NATO countries participating in joint missions (Federated Mission Networking) or the interaction of civilian and military services when providing humanitarian and disaster relief. Federations are often formed on an ad hoc basis, with the main goal of combining forces in a federated mission environment at any time, on short notice, and with optimization of the resources involved [2]. For example, we can cite the situation of a natural disaster, such as an earthquake or flood, that affects a smart city [3]. To coordinate effectively, it is necessary to have as complete and up-to-date a situational picture as possible. A system can be built using data from smart city devices, residents, and the military to provide a complete situational picture obtained in near real-time mode. In such cases, establishing a secure federated IoT environment is required, where different entities can have limited trust relationships and resources already owned by individual organizations are used. The use of federation has several significant advantages, which include simplification of processes, e.g., single authentication, cost reduction through simplification of communication infrastructure, minimization of data storage, efficient data exchange, sharing of resources, etc. When building a data exchange system for a federated IoT environment, we must consider several requirements; the most important are the following.

Enable *authentication and authorization of IoT devices* from various federation organizationsFlexible data management enabling *fine-grained access control* to secure data flow from sources to different recipients.Guarantee the *security and reliability* of data acquisition, flexible processing, and distribution to potential recipients with resource-limited IoT devices.*Decoupling organizations*, i.e., ensuring that no organization can obtain identifying data and the key to communicating with another organization’s devices without its consent; also requires certain conditions defined in the federation’s security policy. No single organization forming a federation can control the entire system.Ensure immediate interoperability (*zero-day interoperability*), that is, readiness for deployment when the need arises for a federated IoT environment.

Meeting the primary requirements of a federated IoT environment is indeed a challenge. It necessitates using an appropriate distributed and multi-component architecture framework, such as a distributed registry and blockchain technology, data dissemination brokers, and security protocols, etc. Our paper primarily focuses on providing authentication of IoT devices and authorization of access to devices and data distributed in the MIoT system. This is one of the main challenges in a federated environment and forms the basis for establishing trust relationships and secure communication between IoT devices belonging to different partners. Ensuring high availability so that MIoT devices can perform their assigned tasks anytime can be achieved by decentralizing and increasing the number of instances of components performing essential security services to create trust between organizations. To meet the requirement of decoupling organizations, we must use data and service dispersion and develop flexible schemes for defining security policies that consider the dynamics of change, including access to IoT devices and the content of transmitted data acquired within an organization. While numerous authentication techniques have been developed and implemented to secure IoT devices, it is important to note that the majority are built on a centralized architecture and rely on a centralized authority, such as a central database or central system services. It is also worth noting that meeting the requirements for IoT device authentication, authorization, and security, or the reliability of services related to the secure distribution of data acquired from different sources (sensors), is much more challenging in a federated environment than in a single-domain environment. The proposed solutions in this work can also be used in a single-domain MIoT environment. Fulfillment of the data-centric security paradigm, i.e., ensuring the secure management of data along the path from its point of origin to the end consumer, combined with the implementation of fine-grained access control mechanisms, can be solved using innovative solutions such as the application of attribute-based encryption (ABE). This novel cryptographic approach enables access control to information based on user attributes, departing from traditional methods reliant on specific keys. First introduced by Sahai and Waters in 2005 [4], ABE embeds access control policies directly in the ciphertext, a departure from conventional encryption models. Despite numerous authors further developing the approach, its practical applications remain relatively unexplored, making it an intriguing area for further research.

This work is a comprehensive effort to develop an architecture framework for an experimental MIoT system. The aim is to securely and reliably distribute various data types in a federation-capable MIoT environment. The experimental system, which authenticates devices based on their IoT device identity (fingerprint), is the results of a thorough research process. The system developed software IoT gateways to provide the verification process by integrating distributed registry technology (Hyperledger Fabric), a message broker (Apache Kafka), data processing microservices (using the Kafka Streams API library), and designed and implemented ABE cryptography data access control methods using lightweight elliptic curve cryptography. The primary objective was to evaluate the performance of the designed attribute-based cryptographic control method and its impact on the latency of data exchange between IoT end devices (sender and receiver). We see our contribution as follows.

Firstly, we built the framework for secure and reliable data dissemination in a federated MIoT environment, enabling IoT device authentication based on their identity (fingerprint), with the use of distributed registry technology (Hyperledger Fabric), a message broker (Apache Kafka), data processing microservices (using the Kafka Streams API library), and software IoT gateways providing the verification process.Secondly, through an appropriate ABE scheme, we introduced a data-centric security paradigm in the experimental MIoT framework that ensures secure data management from the source to the end user.Thirdly, we designed and implemented ABE cryptography data access control methods, using a combination of pairings-based elliptic curve cryptography and lightweight cryptography, and confirmed their suitability for a federated military IoT environment.Fourthly, we built an experimental setup, implemented an access control scheme designed on mobile components (RPi), and performed an evaluation of its performance.

The remainder of the paper is organized as follows. Section 2 reviews related work on blockchain-supported federated networks and ABE. Section 3 briefly characterizes the MIoT architecture and presents the design and specification of requirements for a practical IoT health monitoring system. Section 4 shortly explains ABE’s mathematical foundations. Section 5 presents the system’s overall structure, intended to securely and reliably disseminate data across all command levels and facilitate interoperability. We also highlight our solution for securing information from its inception to its disposal and implementing fine-grained access control mechanisms. In Section 6, we evaluate the feasibility of the proposed solutions by assessing the impact of the applied ABE method on the final delay in the dissemination of the message between the sender and the recipients. Finally, in Section 7 we briefly summarize the results and identify directions for further work on developing the experimental system.

## 2. Related Works

Efforts are underway within NATO countries to integrate data exchange systems between coalition nations [5], resulting in the development of the Federated Mission Networking (FMN) concept. However, one of the significant challenges related to FMN and data dissemination systems is achieving zero-day interoperability. Jansen et al. [6] introduced an experimental MQTT-based environment involving four organizations, distributing data across two configurations. In a study by Suri et al. [7], an analysis and performance evaluation of eight data exchange systems (e.g., RabbitMQ, Redis, DisService) within mobile tactical networks was conducted. De Rango et al. [8] proposed a data exchange system for IoT devices based on the MQTT protocol, with data encrypted using elliptic curves. Additionally, Yang et al. [9] presented a system architecture designed for anonymized data exchange utilizing the federation-as-a-service cloud service model. Moreover, the literature abounds with efforts to integrate the IoT with blockchain technology, also known as distributed ledger technology. The work in [10] delves into both the challenges and advantages of merging blockchain with IoT, emphasizing its implications for data security. Guo et al. [11] put forth a mechanism for authenticating IoT devices across various domains. Their approach leverages distributed ledgers operating in a master–slave configuration for seamless data exchange. Meanwhile, Xu et al. [12] introduced the DIoTA framework, which employs a private Hyperledger Fabric blockchain to safeguard the integrity of data processed by IoT devices. Furthermore, Khalid et al. [13] outlined an access control mechanism for IoT devices, utilizing the Ethereum public blockchain situated in the fog layer, complemented by a public key infrastructure based on elliptic curves. Al-Mekhlafi et al. proposed a scheme for message authentication in 5G networks based on lattices, characterized by high performance and quantum resistance [14]. In [15], Mohammed et al. proposed a device authentication scheme for 5G vehicular networks based on pseudonym authentication. Jarosz et al. designed an authentication and key exchange protocol based on a distributed ledger. Their protocol uses the unique configuration fingerprint of an IoT device. The work by Müller et al. [16] introduces the concept of distributed attribute-based encryption (DABE), where multiple parties can manage attributes and their corresponding secret keys, in contrast to the traditional centralized approach. This distributed model aligns well with the decentralized nature of IoT and blockchain systems. In [17], Jiang et al. propose a blockchain copyright protection scheme based on ABE. This approach aims to address security issues in copyright protection by leveraging the advantages of both blockchain and attribute-based encryption. Lu Ye et al. introduce an access control scheme that combines blockchain and CP-ABE in [18]. This scheme offers fine-grained attribute revocation, keyword search capabilities, and traceability of attribute private keys, enhancing security and flexibility in cloud health systems. Gondalia et al. [19] proposed an IoT-based healthcare monitoring system for war soldiers using machine learning. This system enables army control units to track soldiers’ locations and monitor their health using GPS modules and wireless body area sensor networks (WBASNs). It incorporates various sensors to measure vital signs and environmental conditions. In [20], Sujitha et al. discuss an IoT-based healthcare monitoring and tracking system for soldiers. It emphasizes continuous remote monitoring of troops’ health using IoT technology. Ashish and Kakali propose an edge computing-based secure health monitoring framework for electronic healthcare systems [20]. This framework incorporates ABE techniques to protect medical data and maintain secure monitoring of healthcare information. The research highlights the use of ciphertext-policy ABE (CP-ABE) for data security in IoT-based healthcare systems.

In [21], NIST provided a definition of attribute-based access control (ABAC). ABAC is a logical access control methodology where authorization to perform a set of operations is determined by evaluating attributes associated with the subject, object, requested operations, and, in some cases, environment conditions against policy, rules, or relationships that describe the allowable operations for a given set of attributes. In [22], H. Song et al. discussed a system utilizing an ABAC model deployed on a blockchain, leveraging smart contracts for access control. The system was designed to manage dynamic access control policies effectively, which is crucial in IoT environments where user and resource attributes frequently change. While the aforementioned publications address solutions for data exchange systems, authentication methods, and data access management, a gap remains for a data exchange system that integrates device authentication methods, designed for dynamic and distributed environments, such as the Military Internet of Things, that is also aligned with the data-centric security approach, ensuring data protection from its origin throughout its life-cycle and allowing granular access control.

## 3. Military Internet of Things Environment

### 3.1. Environment Elements

The MIoT environment represents a sophisticated combination of various interconnected devices and systems. Its primary objective is to enhance warfare capabilities, such as situational awareness, logistics, or medical care. The practical military environment comprises numerous devices operating at different levels [23].

At the tactical level, soldiers are equipped with various sensors and communication devices through programs such as the French Army’s FELIN [24], the U.S. Army’s Future Combat Systems [25] and NETT Warrior [26], or the Polish Army’s Tytan.

Mechanized and motorized teams are provided with vehicles, including infantry fighting vehicles (IFVs) or armored personnel carriers (APCs). These vehicles have the potential to offer edge processing capabilities to the team, aligning with the imperative for minimizing the size of soldiers’ equipment. Information from tactical units is transferred to the operational level, where data centers within command structures receive and process it. These data centers also receive data from other sources, such as specialized reconnaissance units or unmanned aerial vehicles (UAVs). In cooperation with other commands and cloud support, they process the information to achieve situational awareness and formulate tasks for subordinate units (Figure 1).

The architecture of the MIoT system is crucial in supporting military operations. It is intricately linked to the specific military scenario and the application domain. In this work, we adopt a general high-level architecture for military applications based on the features and requirements of various military scenarios. The general reference model of the MIoT four-layer architecture is shown in Figure 2.

The *perception and actuation layer (PAL)* can be divided into data acquisition and short-range wireless communication (SRWC) parts and is related to the physical MIoT sensor/actuator nodes. Data acquisition collects information using RFID (radio-frequency identification), sensors, and other technologies. The SRWC concentrates the intercommunication of information on a small scale, for example, in the case of individual dismounted soldier sensors. The primary goal of this layer is to obtain real-time “sensed” data from the environment of each sensor node, including RFID tags/sensors, 5G warfighter phones, Global Positioning System (GPS), devices embedded with sensors, etc. Devices that function as actuators are responsible for carrying out physical actions based on commands received from IoT systems, converting digital signals into operational environments. As part of the MIoT, devices or platforms will include unmanned aerial/ground vehicles, mobile sensors, unattended ground sensors, and individual wearable computer technologies (e.g., sensors that monitor soldiers’ vital functions).

The *communication layer (CL)* transmits the information received from the perception layer to any particular information processing system using existing military communication networks, including access networks (5G, Wi-Fi, etc.) or core networks (Internet/intranet). This layer comprises devices (routers, signal transceivers, etc.) responsible for reliable data transmission. In the military domain, considerable heterogeneity exists among the communication technologies that transmit data between entities. Additionally, non-civilian links such as VHF, HF, UHF, or satellite are used, resulting in network characteristics that strongly differ from conventional wireless communication.

The *data management layer (DML)* effectively processes massive amounts of data and enables intelligent decision making based on network behavior. The core of data management is automated processing technology, which can help implement massive data extraction, classification, filtering, identification, and dissemination.

The *intelligent service layer (ISL)* consists of services comprising this layer, including data storage, computing, data search, system security, and application support. These services would typically be provided by middleware, usually based on the service-oriented approach. In the MIoT, middleware deployment is essential to allow the abstraction of heterogeneous network technologies.

### 3.2. System Requirements

This work presents the design and requirement specifications for a practical soldier-health monitoring system leveraging IoT technology. While focusing on health monitoring, the proposed workflow and architecture can be adapted to various other use cases, such as situational awareness systems or battlefield management systems [27]. The system operates under the following assumptions:Each soldier is equipped with various health monitoring sensors, including but not limited to heartbeat sensors and thermometers.Each sensor is seamlessly connected (paired) to an edge computing device integrated into the soldier’s equipment. These edge devices can perform basic cryptographic operations to ensure data security and integrity.A network infrastructure facilitates the communication between sensors, actuators, and central data centers.Special services are responsible for processing incoming data from sensors, analyzing health metrics, and issuing commands to actuators as necessary.

These assumptions are fundamental for compliance with designed MIoT health monitoring systems, discussed, for example, in [19,20]. Key features of these systems include:Real-time tracking of soldier locations using GPS.Monitoring of vital signs such as body temperature, heart rate, and oxygen levels.Use of wireless body area sensor networks (WBASNs) for data collection.Integration with IoT platforms for data transmission to command centers.Potential incorporation of machine learning for data analysis and prediction.

These systems aim to improve soldier safety, enable rapid response to health emergencies, and enhance overall battlefield awareness for military commanders. By leveraging IoT technology, they provide a more comprehensive and real-time view of soldier health and location compared to traditional methods.

The system should fulfill the following functional requirements regarding security:**End-to-end encryption:** All communication channels between sensors, e.g., devices, and data centers must be encrypted with a security level sufficient for IoT purposes, ensuring that data are protected from unauthorized access or tampering throughout their transmission, safeguarding sensitive health information.**Authentication and access control:** Each device within the network must authenticate itself before participating in data exchange. Access control mechanisms should be enforced to restrict access based on roles and privileges, preventing unauthorized devices from accessing the network, while access control ensures that only authorized users or devices can interact with the system, reducing the risk of data breaches.**Integrity verification:** Data integrity checks should be implemented at each stage of data transmission to detect and prevent tampering or alteration of sensor data, through integrity verification mechanisms, such as cryptographic hash functions, ensuring that data remain unchanged during transit, maintaining their reliability and trustworthiness.**Interoperability in federated environments:** Interoperability within systems, whether military or civilian, is essential for enhancing effectiveness and responsiveness. For instance, it allows health monitoring systems deployed by military units to share real-time health data with civilian healthcare providers, enabling timely medical interventions and resource allocation during joint operations or disaster response efforts. This aspect also gains significance in joint military operations conducted in federated environments involving allied forces. Interoperability extends beyond technical integration to encompass the harmonization of processes, standards, and protocols.**Data-centric security:** With the proliferation of IoT devices and sensors worn by individuals, the volume and variety of health data generated have increased exponentially. Data-centric security focuses on protecting the data rather than solely relying on perimeter defenses, acknowledging that traditional security measures may be insufficient in dynamic and distributed environments. Encryption ensures that data remain unintelligible to unauthorized entities during transmission over the network, mitigating the risk of interception or eavesdropping. Access controls enforce granular permissions, allowing only authorized personnel/systems to access/manipulate health data, reducing the likelihood of data disclosure or breaches.

## 4. Attribute-Based Encryption

Attribute-based encryption (ABE) is a public key encryption paradigm that provides fine-grained access control. In ABE, both the user’s secret key and the ciphertext are associated with attributes that determine the ability to decrypt the ciphertext. This is achieved through the use of pairing-friendly curves and bilinear maps. Both KP-ABE and CP-ABE offer different approaches to policy enforcement, catering to various application requirements in secure data sharing.

There are two main types of ABE: key-policy attribute-based encryption (KP-ABE) and ciphertext-policy attribute-based encryption (CP-ABE). In KP-ABE, the ciphertext is associated with a set of attributes, and the user’s secret key is associated with an access policy. The key policy (KP-ABE) determines which ciphertexts the user can decrypt. In contrast, in CP-ABE, the user’s secret key is associated with a set of attributes, and the ciphertext is associated with an access policy. In the scope of this study, CP-ABE is used for the reasons specified in Section 5.3.

### 4.1. Bilinear Pairings

Attribute-based encryption schemes often rely on bilinear pairings of elliptic curves, known as pairing-friendly curves. Let $G_{1}$ and $G_{2}$ be cyclic groups of prime order *p*, and let $e:G_{1}\times G_{1}\to G_{2}$ be a bilinear map. The properties of the bilinear map are the following:**Bilinearity:** $e\left(aP,bQ\right)=e{\left(P,Q\right)}^{ab}$ for all $P,Q\in G_{1}$, and $a,b\in Z_{p}$.**Non-degeneracy:** e(g,g)≠1 if *g* is a generator of $G_{1}$.**Computability:** There exists an efficient algorithm to compute e(P,Q) for all $P,Q\in G_{1}$.

### 4.2. ABE Procedures

The CP-ABE scheme can be formally split into following procedures:**Setup**(λ): This procedure initializes the system.
(a)Choose bilinear groups G and $G_{T}$ of prime order *p*. These groups are fundamental to cryptographic operations.(b)Select a generator g∈G. This will be used to generate other group elements.(c)Choose random exponents $\alpha ,\beta \in Z_{p}$. These serve as the core secrets of the system.(d)Compute the public parameters:
$h=g^{\beta }$: This hides β while allowing its use in computations.$f=g^{1/\beta }$: This is used in encryption and decryption.$Y=e{\left(g,g\right)}^{\alpha }$: This is a pairing operation that hides α.(e)Set $\mathrm{PP}=(G,G_{T},p,g,h,f,Y)$ as the public parameters.(f)Set $\mathrm{MSK}=(\beta ,g^{\alpha })$ as the master secret key.(g)Output PP and MSK. PP is made public, while MSK is kept secret.**KeyGen**(MSK, *S*): This procedure generates a secret key for an attribute set *S*.
(a)Choose a random $r\in Z_{p}$. This randomizes the key for security.(b)Compute $D=g^{(\alpha +r)/\beta }$. This embeds the master secret into the key.(c)For each attribute i∈S:
Choose a random $r_{i}\in Z_{p}$. This further randomizes each attribute component.Compute $D_{i}=g^{r}\cdot H{\left(i\right)}^{r_{i}}$. H(i) is a hash function mapping attributes to group elements.Compute $D_{i}^{\prime }=g^{r_{i}}$. This allows for cancellation in decryption.(d)Set ${\mathrm{SK}}_{S}=\left(D,{\left\{\left(D_{i},D_{i}^{\prime }\right)\right\}}_{i\in S}\right)$ as the secret key for attribute set *S*.(e)Output ${\mathrm{SK}}_{S}$.**Encrypt**(PP, *m*, A): This procedure encrypts a message *m* under an access policy A.
(a)Choose a random $s\in Z_{p}$. This randomizes the encryption.(b)Compute $\tilde{C}=m\cdot Y^{s}=m\cdot e{\left(g,g\right)}^{\alpha s}$. This hides the message with the master secret.(c)Compute $C=h^{s}$. This is used in decryption to recover the message.(d)For each attribute *i* in the access structure A:
Choose a random $s_{i}\in Z_{p}$. This randomizes each attribute in the policy.Compute $C_{i}=g^{s_{i}}$.Compute $C_{i}^{\prime }={\left(g^{s}\cdot H{\left(i\right)}^{-s_{i}}\right)}^{1/\beta }$. This embeds the policy into the ciphertext.(e)Set $\mathrm{CT}=(A,\tilde{C},C,{\left\{\left(C_{i},C_{i}^{\prime }\right)\right\}}_{i\in A})$ as the ciphertext.(f)Output CT.**Decrypt**(PP, CT, ${\mathrm{SK}}_{S}$): This procedure attempts to decrypt a ciphertext using a secret key.
(a)First, check if the attribute set *S* satisfies the access policy A. If not, output ⊥ (decryption failure).(b)If *S* satisfies A:
Compute $e{\left(g,g\right)}^{\alpha s}$ using CT and ${\mathrm{SK}}_{S}$:
This involves pairing operations and computations based on the satisfying set of attributes.The computation cancels out randomizing factors, leaving only $e{\left(g,g\right)}^{\alpha s}$.Recover the message: $m=\tilde{C}/e{\left(g,g\right)}^{\alpha s}$.Output the decrypted message *m*.

In the decryption process of ciphertext-policy attribute-based encryption (CP-ABE), recovering $e{\left(g,g\right)}^{\alpha s}$ is a crucial step. The decryption algorithm uses the ciphertext CT and the secret key ${\mathrm{SK}}_{S}$ associated with an attribute set *S* that satisfies the access policy A. The decryption process involves pairing operations and computations using these components. The goal is to cancel out all the randomizing factors, leaving only $e{\left(g,g\right)}^{\alpha s}$. In simplified form:$$\frac{e(C,D)}{\prod_{i\in I}\frac{e(C_{i},D_{i})}{e(C_{i}^{\prime },D_{i}^{\prime })}}=\frac{e(h^{s},g^{(\alpha +r)/\beta })}{e{\left(g,g\right)}^{rs}}=\frac{e(g^{\beta s},g^{(\alpha +r)/\beta })}{e{\left(g,g\right)}^{rs}}=\frac{e{\left(g,g\right)}^{\alpha s}\cdot e{\left(g,g\right)}^{rs}}{e{\left(g,g\right)}^{rs}}=e{\left(g,g\right)}^{\alpha s}$$
where *I* is the set of attributes used in the decryption and $\frac{e(C_{i},D_{i})}{e(C_{i}^{\prime },D_{i}^{\prime })}$ is a term designed to equal $e{\left(g,g\right)}^{rs}$.

The exact computation depends on the specific CP-ABE scheme used, but the general idea is to use the properties of bilinear pairings and the structure of the keys and ciphertext to isolate $e{\left(g,g\right)}^{\alpha s}$.

## 5. Data Exchange System

The following headings provide an overview of our multilayered experimental system for IoT-based environments and explain the rationale for implementing a data-centric security approach within our architecture.

### 5.1. Experimental Environment Architecture

Figure 3 depicts the overall structure of the system, which is intended to securely and reliably disseminate data across all command levels and facilitate interoperability among different entities, including sensors, actuators, and data centers, from various organizations. The environment (Figure 3) illustrates a federation of two organizations (Org1, Org2) and consists of several layers that perform the following functions:**The publishers layer** represents authenticated and authorized entities (sensors and actuators) that produce and secure messages through the sealing process. The device’s fingerprint (identity) is the key used for the sealing process.**The subscribers layer** comprises authenticated and authorized entities that read available data from the Kafka cluster layer.**The Kafka cluster layer** is compromised of Apache Kafka message brokers that acquire, merge, store, and replicate data generated from the publishers layer (producers) and make it available to the subscribers layer (consumers). Apache Kafka operates on a producer–broker–consumer (publish–subscribe) model, facilitating the classification of messages based on their respective topics. The inherent synchronization mechanisms and distributed data replication among brokers ensure the continuous availability and reliability of data records. Furthermore, Kafka’s serialization and compression techniques (e.g., lz4, gzip) enable the system to remain agnostic to data formats and network protocols, thereby ensuring compatibility and robustness in heterogeneous environments.**The Streams microservice layer** is primarily utilized to verify sealed messages. Additionally, it can be used to analyze, group, and share messages related to relevant entities and enrich them (e.g., by detecting objects during image processing). The system leverages the built-in primitives of the Kafka cluster layer, such as failover and fault tolerance. Additionally, it employs a semantic guarantee pattern ensuring that each record (message) is processed exactly once end-to-end. Consequently, even in the event of a stream processor (microservice) failure, records are neither lost nor processed multiple times. In the proposed system, it is also utilized for ABE re-encryption.**The device maintain layer** manages (e.g., define, register, retire) device identity images stored in the distributed ledger. In the proposed system, it is also responsible for ABE attribute and device management.**The communication layer** enables the Streams microservice and the device maintain layer to communicate with the distributed ledger layer via a hardware–software IoT gateway. Moreover, a dynamic mode is proposed for the connection profile. This profile utilizes the ledger nodes’ built-in mechanism to continuously detect changes in network topology. Consequently, microservices will be able to operate reliably, even in the event of some node failures.**The distributed ledger layer** redundantly stores the identities of devices belonging to organizations participating in the federation. A permissioned blockchain that employs the Practical Byzantine Fault Tolerance (PBFT) consensus protocol is proposed. In protocols of this nature, all participants must be mutually known, necessitating the use of a public key infrastructure (certificates) for identity verification. The execution of complex business logic, such as device registry, is facilitated by invoking multilingual chaincode (Go, Java, Node.js). Chaincode implements a collection of smart contracts (transaction steps) and defines an endorsement policy, specifying which organizations must authorize a transaction.

The article by Zieliski et al. [28] discusses the rationale behind the use of components within the system layers, which is not covered in this publication. The following section outlines the flow of messages and the specific actions taken within our system, with Figure 4 providing a detailed illustration of our experimental system.

Within our system, entities representing the publishers layer secure their messages by sealing (tagging) them with registered identity. These sealed messages are then sent using the available communication protocol and medium to the Kafka cluster layer, where they will be stored on a specific topic, such as “cctv-1-in”. The next step involves the Streams microservice layer, which reads messages from brokers belonging to the Kafka cluster layer to verify sealed messages.

During verification, the microservice queries the Hyperledger Fabric ledger for an image of the device identity through the IoT gateway (communication layer). This query aims to compare the identity extracted from the message with the one stored in the distributed ledger layer. Once a message is verified and approved, it is written again on the Kafka cluster layer within a dedicated topic (e.g., cctv-1-out) and provided to entities representing the subscribers layer.

### 5.2. Authorization Challenges

The Apache Kafka technology has built-in components and mechanisms to define and handle entity (device) authorization using access control lists (ACLs). In a nutshell, ACLs specify which entities can access a specified resource and the operations they can perform on that resource. One of the specific types of resource is a Kafka topic, which organizes (stores) messages produced by the producers layer. The ACL operation depends on the resource type, and in the context of our solution, the READ and WRITE operations assigned to the entity (principal) for the topic are relevant.

Establishing a separate principal for each entity (device) and assigning only the necessary ACLs streamlines debugging and auditing, as it identifies which entity is performing each operation. For instance, within an environment with three devices, three principals should be created: device_job_01, device_job_02, and device_job_03. Afterward, for each principal, only the essential operations should be allowed, and each should be used with its designated principal. Kafka ACLs are defined in the general format of *“Principal P is [Allowed/Denied] Operation O From Host H On Resources matching Resource/Pattern RP”*. An example ACL rule definition is presented below:


kafka-acls --bootstrap-server localhost:9092



  --command-config configs.conf  --add



  --allow-principal User:uav_1 --allow-host



  192.168.1.13 --operation write  --topic cctv-1-in


It means that the principal (user) named uav_1 is allowed to perform a WRITE operation on the topic cctv-1-in from IP address 192.168.1.13. In this example, a basic plain text principal was used. For real-world use case scenarios, protocols such as the SASL security protocol with GSSAPI (Kerberos) or TLS are more likely to be utilized. This implies that different security (authentication) protocols require different formats of the principal:


--deny-principal User: CN=uav_2, OU=Org2, O=None, C=None


Moreover, large Kafka cluster topologies with numerous Kafka partitions or many publishing/subscribing entities can face significant challenges in managing entity authorization. Implementing a more complex authorization hierarchy in the Kafka cluster layer is possible, but it also comes with operational overhead that requires the following:Deployment of an external server that associates users with roles and/or groups (e.g., LDAP server);Definition of a detailed ACL by specifying permission type for group, role, and user (ACL stored in Kafka cluster layer);Integrating a custom implementation of authorizer with the user association server.

### 5.3. Attribute-Based Encryption Application

One of the possible solutions to the mentioned challenges is the application of ABE in our experimental environment. This can provide more fine-grained access control than the Kafka cluster layer. Using ABE ACLs in our solution enables dynamic authorization at the individual message level. Additionally, this can relieve the Kafka cluster layer from the burden of the authorization process, allowing its internal resources to be released, and the Streams microservice layer can handle access control. In addition, operational costs are low and the ABE can be implemented in environments of any scale that involve multiple participating organizations. Finally, entities within organizations (such as the publishers/subscribers layer) can maintain the ability to move freely between Kafka cluster layers in different organizations.

Recent ABE schemes are tailored to address specific requirements, showing variability in their capabilities, as highlighted in [29]. In the scope of this study, we opted for the scheme introduced by [30], mainly due to its provision of:Unbounded attribute sets and policies: The algorithms can handle attribute sets and policies of any size without predetermined limits.Support for negation and multi-use of attributes: This allows for more complex and expressive access control policies.Fast decryption: The schemes are designed to perform decryption operations quickly, enhancing overall system performance.Full security under standard assumptions: The algorithms provide strong security guarantees based on well-established cryptographic assumptions.

This choice aligns with the architecture of Kafka’s topics, for which attributes represented in a key:value pattern are easily processed. The necessity for repeated attributes in this pattern within Kafka’s topic architecture underscores the compatibility and appropriateness of the selected ABE scheme for the application of our system for the soldier-health monitoring system.

### 5.4. Design Proposition

#### 5.4.1. General Overview

The data-centric security paradigm necessitates secure handling of information from its inception to its disposal, coupled with the implementation of fine-grained access control mechanisms. While ABE addresses these imperatives, it imposes substantial computational and bandwidth requirements. This study advocates a paradigm shift, since the sensors and edge processing layers predominantly comprise devices with limited computational capabilities. Specifically, we propose a scheme in which the responsibility for ABE encryption is delegated to the organizational infrastructure, presumed to possess the requisite cryptographic capabilities. Concurrently, sensors and it is envisaged that their paired edge processing devices will execute less computationally intensive cryptographic operations, such as AES encryption, SHA256 hashing, and HMAC-SHA256.

In the proposed framework (Figure 5), sensors employ symmetric encryption to encapsulate the generated data. Although the choice of encryption algorithm is not the focal point of this study, it prompts discussion on the potential applicability of lightweight cryptography. Devices with considerable computational prowess, such as Raspberry Pi or smartphones, predominantly utilize AES encryption. However, encryption and HMAC keys are derived from a sensor-specific fingerprint using a predefined key-derivation function. The sensor divides the generated message into two distinct components:**Encrypted data segment:** Encrypted with key derived from the device fingerprint; it consists of (1) encrypted message, (2) encrypted ABE policy, (3) unencrypted session GUID.**Preamble:** Used to authenticate message; it consists of (1) HMAC of encrypted data segment (with key derived from fingerprint), (2) device GUID, (3) session GUID.

Subsequently, these components are published in Kafka input topics. Incoming messages undergo processing through the following sequence of operations within Kafka microservices: 1. Retrieval of the sensor fingerprint from Hyperledger using the provided device GUID, 2. key derivation for encryption and HMAC, 3. HMAC verification, 4. message decryption, 5. attribute-based re-encryption based on the policy encapsulated within the main segment, and 6. publication of the ABE-encrypted message on the topic specified in the policy. Microservices require at least one of the attributes to be a topic attribute. This ensures that topics associated with data recipients remain unencumbered.

The aforementioned mechanism is exclusively employed for streaming-type connections to distribute the session key associated with the session GUID. Subsequent messages comprise solely the message encrypted with the session key and the session GUID.

#### 5.4.2. Attribute-Based Encryption Message Flow

Consider the transmission scenario of sending a report from a sensor:The sensor generates data; for example, a smartphone equipped with a heart rate sensor detects an anomaly in heart activity. This information is appended with an attribute logic sentence and encrypted using the device fingerprint. Let the logic sentence be MEDICAL and GRID28B, where GRID28B represents the sector number obtained from the location sensor, also associated with this smartphone. GUIDs and HMAC are further appended as delineated in the system overview.The sensor sends the data to a Kafka input topic, where processing is executed as described in the system overview.Devices subscribed to the MEDICAL topic receive the message. If they satisfy the attached attribute logic sentence, they decrypt the message using their current keys at the time of data transmission. The Kafka microservice preliminarily verifies the authenticity of the message.

#### 5.4.3. Attribute-Based Encryption System Setup

The initialization of the ABE system encompasses the following sequential steps (Figure 6) and is compliant with the secure onboarding process described in [31]:Establishment of ABE system parameters, including the master secret key.Pairing data recipients possessing limited computational capabilities with more robust devices by sharing a common symmetric key for communication, treating them as a unified entity.Loading ABE system parameters onto the designated devices intended for system operation.Associate private keys with device identifiers ID on devices designated for data reception. An identifier is any piece of information used to uniquely distinguish a device, such as a device GUID generated in this step or a combination of type and number, e.g., UAV-1. These keys enable the distribution of attribute keys for specific devices when granting new permissions or authenticating attribute requests. This process is recorded in Hyperledger as device registration.Loading private keys corresponding to granted attributes onto the respective devices. This action is documented in Hyperledger as attribute granting.

#### 5.4.4. Attribute and Device Revocation

A significant challenge associated with ABE systems is related to the revocation of attributes or devices. While public key infrastructure (PKI) solutions mitigate this issue via certificate revocation lists, ABE lacks such certificates. Consequently, ABE commonly employs mechanisms such as timestamps for attribute or device management. However, the integration of Hyperledger allows for an alternative approach, associating attributes with the block number of the most recent revocation. Consider the following scenario (Figure 7):A specific attribute, such as MEDICAL, is granted to any number of devices, with each grant action recorded in Hyperledger.Upon revocation of the recipient attribute, this incident is documented in Hyperledger.New attributes associated with a revocation block number, e.g., UAV-123, are issued to all other devices. Distribution occurs by encrypting new private keys with each device’s identity and publishing them to the Kafka device management topic.The microservice responsible for ABE re-encryption employs the new attribute public key.

If, instead of an attribute, a device is revoked from the system, KGC has to issue new keys for all attributes that the device possesses to other recipients. Additionally, compromised data generators can be reported in Hyperledger to ignore messages originating from them. This can be facilitated through device GUID revocation. It is crucial to recognize that distributing new keys through a Kafka topic mitigates any potential data availability gaps. These keys become accessible concurrently with the encrypted data in the topics, ensuring seamless data access.

#### 5.4.5. Granting New Permissions

Consider a scenario of granting new permissions (Figure 8):The recipient device sends a request to KGC, signing it with its identifier.In KGC, after signature verification, a decision is made to grant a new key. It is reported on Hyperledger and sent back to the device, signed by KGC, and encrypted with the recipient identifier.After signature verification and decryption, if an attribute is of a topic type, the device subscribes to the new topic in the Kafka broker.

### 5.5. Vertical Industry Usage

Vertical industries are diverse and their stringent requirements are dictated by the service characteristics of the related vertical segment, hence providing an vertical industry-optimized platform is a complex issue, especially for performance (e.g., latency) and functional requirements (e.g., access management).

The long-term performance of a specific vertical industry can be assessed from a business perspective using key performance indicators (KPIs), e.g., latency, connection density, and wide-area coverage. These KPIs help identify the most suitable primary use case class for the vertical industry within 5G networks. The mentioned classes related to 5G technology can be categorized into Enhanced Mobile Broadband (eMBB), Massive Machine Type Communication (mMTC), and Ultra-Reliable and Low-Latency Communication (URLLC). The eMBB class requires high throughput and mobility. For example, non-stationary IoT devices that produce audiovisual data streams need to be processed in a short time. However, the mMTC class has specific design requirements for federated environments, while URLLC relies on reliable, low-latency connections crucial in e-health.

Taking into account the above, network slicing, a key feature of 5G, enables addressing the specific requirements of vertical industries by deploying particular industry system architectures in a versatile manner and integrating them with our data exchange system (scheme). This is achieved while running multiple concurrent verticals on a shared infrastructure and reducing operational costs. Additionally, within network slicing, a granular and customizable network isolation mechanism enables the creation of different slice configurations (full/partial and logical/physical) that restrict industry resources from being accessed by other entities.

The following presents a general proposal for how our data exchange system (security scheme) will work in several vertical industries implemented in parallel network slices (Figure 9).

For the Kafka cluster layer, it is possible to use various isolation configurations. This allows the cluster (brokers) to be shared among different vertical industries (slices) due to the logical separation via Kafka topics. Additionally, the Kafka architecture is data independent, which allows for data protection (isolation) with different encryption and authorization mechanisms (e.g., ABE). Therefore, we propose the deployment of Kafka clusters among industry premises (cooperating parties).

Numerous considerations exist in the literature on the integration of distributed ledger technology with 5G networks. An example is the work of Praveen et al. [32], which presents a model that enables multi-operator 5G infrastructure sharing, where information associated with a customer’s SIM card is stored in a distributed ledger. Salahdine et al. [33], in addition to studying classes of attacks on network slicing technology, highlight the importance of distributed ledgers in the context of 5G. We propose integrating the distributed ledger layer into the scope of the 5G core network. In this configuration, the realm ledger is shared among slices owned by partners in the same vertical industry. It is worth highlighting that while this configuration is not obligatory, an alternative scenario could involve a single main ledger across slices, accomplished by setting up different channels of the Hyperledger blockchain.

The microservice pools (virtualized execution environments) executing the ABE algorithm can be tailored based on the needs of the associated network slices for resource provisioning and allocation. Moreover, our scheme allows for sharing a microservice pool between slices that operate on the same distributed ledger.

## 6. Experimental Results

### 6.1. Experiment Objective

The primary objective of the experiments conducted on the described system is to ascertain its feasibility for deployment in real-world conditions. To investigate this, the following metrics were employed, providing a framework for evaluating the system’s performance, security, and scalability in practical scenarios:**Encryption and decryption time using ABE:** Evaluates the time required to encrypt and decrypt data using ABE.**Latency in data transmission via Kafka and device verification in Hyperledger**: Measures the delay associated with transmitting data through the Kafka stream broker and the subsequent retrieval and verification of data within the Hyperledger framework.**Key generation time:** Examines the time taken to generate new encryption keys when new keys are issued or existing keys are revoked.**Size of the resulting ciphertexts:** Assesses the size of the encrypted data produced through the ABE processing.

### 6.2. MIoT Context

In the experiments, efforts were made to replicate the operational conditions experienced by units of various levels on the battlefield. Given the assumptions and requirements outlined in Section 3.2, we assume that soldiers on the battlefield are equipped with medical monitors (data generators) paired with devices possessing sufficient computational power to aggregate signals from the soldiers’ medical sensors and perform basic cryptographic operations (AES, HMAC), as described in Section 5.4, as well as decrypt ABE ciphertexts (for cases where the soldier is both the producer and consumer of data). For the purposes of the experiment, this device is represented by a Raspberry Pi 4 or 5. The organization that operates the medical system has its data center at the command post of the corresponding unit, which contains a Kafka broker, Hyperledger Fabric blockchain, and microservices. This data center is situated in a location such as a container mounted on a truck (e.g., Polish AWRS) and has sufficient space to accommodate larger devices. For the experiment, these devices are represented by the x86 platform, as well as clusters of smaller devices like the Raspberry Pi 5. The devices within the container are interconnected via wired connections, while wireless connections are used with other devices. Data recipients are soldiers on the battlefield, equipped with devices capable of performing ABE decryption, represented by Raspberry Pi 4/5, as well as other data centers, and command apparatuses represented by the x86 platform.

### 6.3. Implementation

The described environment and data exchange system have been realized in a simplified setup, comprising a data generator, a data recipient, and a Kafka broker. The Kafka broker is deployed within two Raspberry Pi 5 clusters as three instances (two instances on one of the devices and one on the other) with Hyperledger and the ABE re-encryption microservice interconnected through Kafka Streams mechanisms. Hyperledger is deployed on an x86 machine, while the re-encryption service is deployed interchangeably on a Raspberry Pi 4, Raspberry 5 or x86 machine. The data generator produces and encrypts messages as described in the previous section, while the data recipient subscribes to the output topic within the Kafka broker. The data generator and the recipient are implemented as Go scripts. The data generator, re-encryption service, and data recipient scripts are available in [34]. For ABE, the CIRCL cryptographic library by Cloudflare [35], implementing [30], is integrated. The elliptic curve used is BLS12-381 described in [36,37], providing 126 bits of security [38]. BLS12-381 belongs to the Barreto–Lynn–Scott (BLS) family of curves, with the number 12 indicating its embedding degree and number 381 referring to the size of its prime field: a 381-bit prime. This curve operates over a prime field $F_{p}$, where *p* is a 381-bit prime:$$\begin{array}{cc} p= & 0x1a0111ea397fe69a4b1ba7b6434bacd7\\ \; & 64774b84f38512bf6730d2a0f6b0f624\\ \; & 1eabfffeb153ffffb9feffffffffaaab \end{array}$$
and features a curve order *q* that is a 255-bit prime:$$\begin{array}{cc} q= & 0x73eda753299d7d483339d80809a1d805\\ \; & 53bda402fffe5bfeffffffff00000001. \end{array}$$

It defines three groups: $G_{1}$, a subgroup of points on the curve $E\left(F_{p}\right):y^{2}=x^{3}+4$; $G_{2}$, a subgroup of points on the twist curve $E^{\prime }\left(F_{p^{2}}\right):y^{2}=x^{3}+4\left(u+1\right)$, where $F_{p^{2}}=F_{p}\left[u\right]/\left(u^{2}+1\right)$; and $G_{T}$, the target group for pairing operations. This structure supports an efficiently computable bilinear pairing $e:G_{1}\times G_{2}\to G_{T}$. BLS12-381 is particularly useful in scenarios requiring pairing-based cryptography, offering a good trade-off between security and performance for modern cryptographic applications.

The logical scheme of the simplified setup is presented in Figure 10.

### 6.4. Performance Evaluation

The flow of messages from the generator to the recipient was emulated using the following methodology. The time taken for ABE encryption and decryption of 32-byte messages was measured in varying numbers of attributes, along with the length of the ciphertext for 1000 messages. The results are presented in Table 1 and Table 2. Extreme values were excluded in each quantile order (p0.99, p0.95, and p0.9). The emulation was executed on a PC equipped with an x86 13th Gen Intel(R) Core(TM) i5-13600KF processor operating at 3.50 GHz. Furthermore, performance tests were conducted on a Raspberry Pi 4 and Raspberry Pi 5 using the same number of attributes as in the PC test. Both the data generator and the recipient were executed on the Raspbian OS as Golang scripts.

The results indicate that the proposed scheme is viable for the number of attributes typically assumed to be used in battlefield networks. For scenarios involving eight attributes, the processing time for encryption and decryption increases by approximately 100 ms, even when all messages are encrypted using ABE. Using session keys that incorporate ABE introduces only a minimal increase in the delay in connection establishment. The computational time of AES is assumed to be negligible.

While there is a significant overhead associated with the ciphertext size, this challenge has been effectively addressed through the adoption of hybrid cryptography. In this framework, ABE is utilized primarily for session key encryption and distribution. For individual messages, ABE can be employed to encrypt only the symmetric key.

Nevertheless, ciphertexts of approximately 4 KB can still present challenges for IoT networks that rely on wireless protocols. To transmit around 4000 bytes of ciphertext over networks with limited payload sizes, the data must be fragmented into multiple packets. This fragmentation not only introduces additional overhead but also complicates the communication process, leading to increased transmission times and higher energy consumption. For instance, the maximum payload size for LoRaWAN is 242 bytes, with each packet consisting of an additional 13 bytes for headers and other protocol-specific information [39]. In comparison, the widely used cryptographic scheme RSA-3072, similar to BLS12-381, believed to provide approximately 128 bits of security, generates ciphertexts of around 384 bytes.

The results presented in Table 3 and Table 4 indicate that ABE encryption may not be feasible for devices with limited computational capabilities. In contrast, ABE decryption requires considerably less time, which supports the strategy of managing ABE re-encryption within the Kafka microservice and executing only decryption on the recipient devices.

In [28], it was demonstrated that the processing times for Kafka deployed in AWS averaged approximately 40 ms per message (with IdentityCount=100,000). This processing involved receiving messages on a topic, verifying the data generator in the blockchain, and forwarding the data to an output topic. In this experimental setup, the average time of Kafka responding to requests was 17.3 ms and the average time of verifying the data generator in the blockchain was 31.3 ms with IdentityCount=10,000, summing up to an average time of 48.6 ms of additional processing of the packet between it appearing in the broker and being received by the consumer. The time for conducting other operations, such as calculation of HMAC and AES decryption, is negligible in comparison. The combined processing times with ABE re-encryption are detailed in Table 5 and Table 6.

It is important to note that, provided the Kafka broker, blockchain, and microservices possess adequate computational power and the network is not congested, the disadvantage of employing ABE for symmetric key distribution is the latency associated with establishing the streaming session, but this does not impact the continuous processing of the data stream. The observed latency for establishing a streaming session is less than 0.5 s for eight attributes on both Raspberry Pi 4 and 5.

In Table 7, the average times for new key generation as a function of the number of attributes are presented. The data indicate that delegating these computations to devices with constrained computational capabilities is impractical for large networks. Within the proposed system, while key revocation is limited to the number of devices (O(n)), the occurrence of a revocation necessitates the re-generation of keys for all other devices issued with the same key ($O(n^{2})$). If the number of devices in the network is large, it may be necessary to further accelerate computations, for example, by parallelizing techniques using GPUs.

### 6.5. PKI Comparison

As an alternative to attribute-based encryption (ABE), an attribute-based access control (ABAC) approach integrated with a public key infrastructure (PKI) can be employed. In this approach, the microservice responsible for re-encryption would not rely on ABE policies but instead determine the recipient list based on issued attributes and distribute messages using PKI methods. This re-encryption microservice would effectively combine the roles of the policy decision point (PDP) and the policy enforcement point (PEP), as shown in [21]. Unlike traditional ABAC, where a subject initiates a request for data access and the PDP decides whether to grant it, this mechanism differs by proactively determining recipients and distributing data, akin to the broadcasting model in the ABE approach discussed in this work. However, it is feasible to propose that the re-encryption service, acting as the PDP, could preemptively decide on the recipient list and distribute messages via Kafka topics. This could be achieved by publishing the messages using each recipient’s public key, stored along with issued attributes on a blockchain. Such a strategy would make the data accessible in a manner similar to the ABE-based approach. This model is equivalent in terms of cryptographic operations to the classical authentication in PDP by the entity, because such authentication is performed through a digital signature (RSA) using a certificate. The cryptographic operations of signing and encryption are the same for the RSA algorithm (signing with the private key and verification with the public key).

To investigate this alternative, a supplementary benchmark was conducted in addition to the previously described experimental setup. This benchmark aimed to compare the performance of the TKN20 scheme, based on the BLS 12-381 elliptic curve, against RSA algorithms. RSA was chosen for its widespread adoption as a key wrapping mechanism in PKI systems. Various RSA key lengths (2048, 3072, and 4096 bits) were considered to ensure a comprehensive evaluation.

For the TKN20 scheme, two configurations were assessed: one using a two-attribute policy (referred to as “short”) and the other employing an eight-attribute policy (referred to as “long”). The benchmark measured the execution times of key generation, encryption, and decryption processes for each configuration.

The benchmark tests were conducted on a PC equipped with a 13th Generation Intel(R) Core(TM) i5-13600KF processor, operating at 3.50 GHz. Each algorithm was executed 1000 times to ensure statistical significance. The results are presented in Table 8.

Given that RSA public keys are optimized for rapid computations (leveraging Fermat’s numbers), the encryption times for RSA were significantly faster. Consequently, the comparison focused on the combined times for both encryption and decryption. When operating at a security level comparable to BLS 12-381, RSA-3072 was found to be approximately 22 times faster for encrypting and decrypting messages with a 2-attribute policy and about 30 times faster for messages with a 50-attribute policy. These findings suggest that the TKN20 scheme could be viable for deployment in large-scale networks, particularly when the number of potential data recipients exceeds the ratios identified in the comparison. This advantage is further amplified when considering that a re-encrypting microservice would need to determine the recipient list based on its own access control analysis and also considering that in ABE only one encryption per message is conducted, as opposed to RSA with encryptions for each recipient.

In Table 9, ciphertext overheads are presented. The overhead ratios (8.4 and 9.8) further indicate that it may be more beneficial to use ABE instead, particularly when efficiency and overhead are critical considerations.

MIoT networks are typically large-scale systems. Given that each soldier is equipped with multiple devices, the number of potential data recipients increases exponentially depending on the operational level (platoons, companies, battalions, etc.). Attribute-based encryption (ABE) cryptography has the potential to significantly outperform traditional public key infrastructure (PKI) solutions in such environments. Moreover, ABE aligns with the core principles of data-centric security (DCS), an approach currently being developed and implemented not only by NATO and its member states, including Poland, but also by major corporations such as Orange. These principles include fine-grained access control, data protection across boundaries, scalability and flexibility, and policy enforcement at the data level.

## 7. Conclusions

In this study, we introduced a comprehensive data exchange system that includes all of our desired functionalities. Our approach combines ABE with a blockchain-enabled device management system. This combination not only alleviates challenges associated with ABE, such as computational overhead, but also retains its inherent strengths, including distribution properties and the ease of adopting a data-centric security approach, compared to traditional PKI systems.

Our next steps will involve implementing and deploying this system in a real-world setting. While we have estimated delay times resulting from ABE calculations, deploying the system in a real environment will provide more accurate insights, such as network latency considerations.

Additionally, we are intrigued by the prospect of evaluating ABE calculation times on different less powerful devices commonly used in IoT and using lightweight libraries and implementations. This exploration could offer valuable insights into the system’s scalability and performance on resource-constrained platforms.

It should be emphasized that the proposed solution establishes a security layer within information processing systems (e.g., in the data-centric security approach), where security should be evaluated as a complex whole. Further work in this direction is planned for subsequent steps.

## Figures and Tables

**Figure 1 sensors-24-05863-f001:**
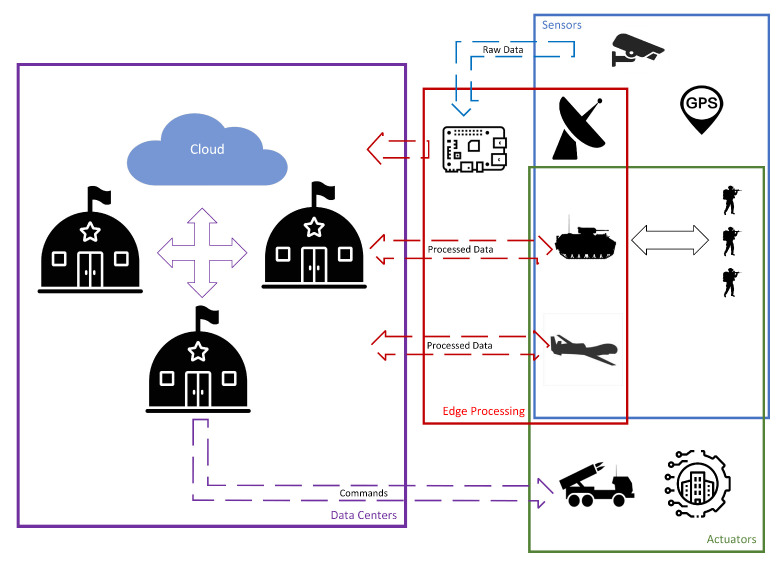
High-level scheme of MIoT main components.

**Figure 2 sensors-24-05863-f002:**
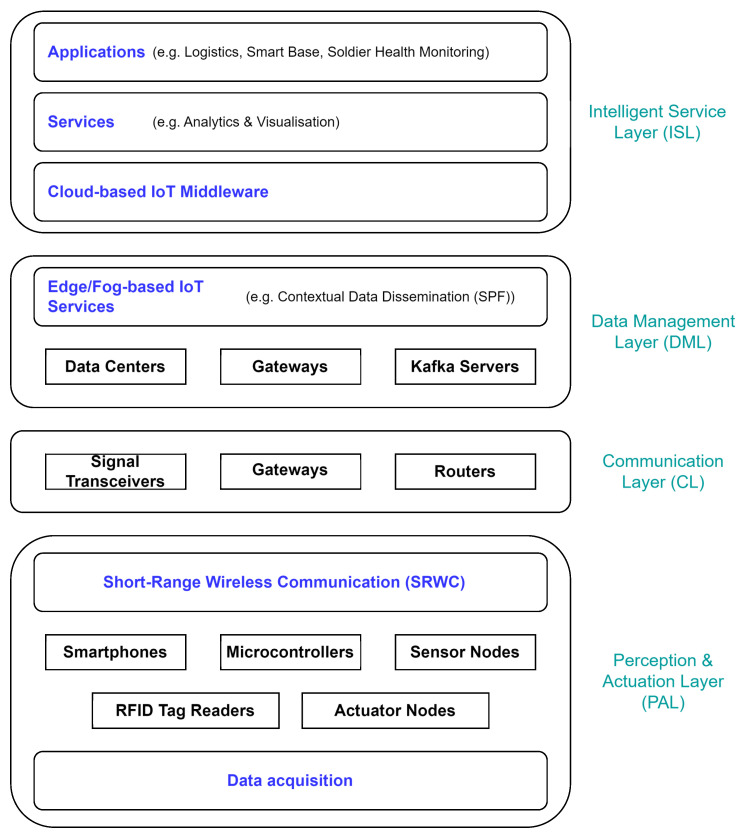
The MIoT layered architecture.

**Figure 3 sensors-24-05863-f003:**
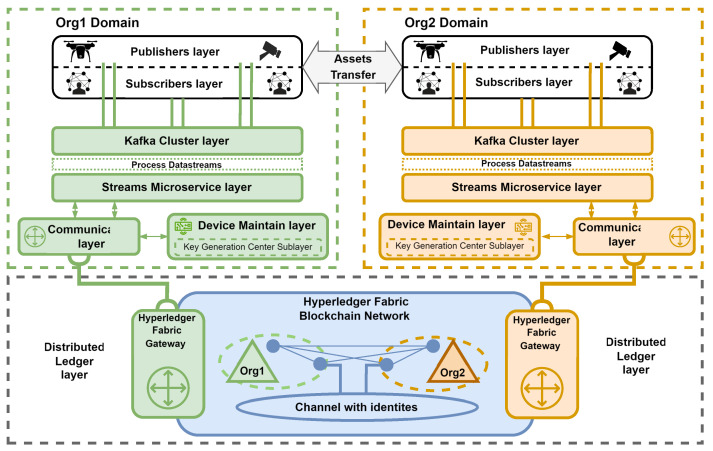
General overview of the experimental environment.

**Figure 4 sensors-24-05863-f004:**
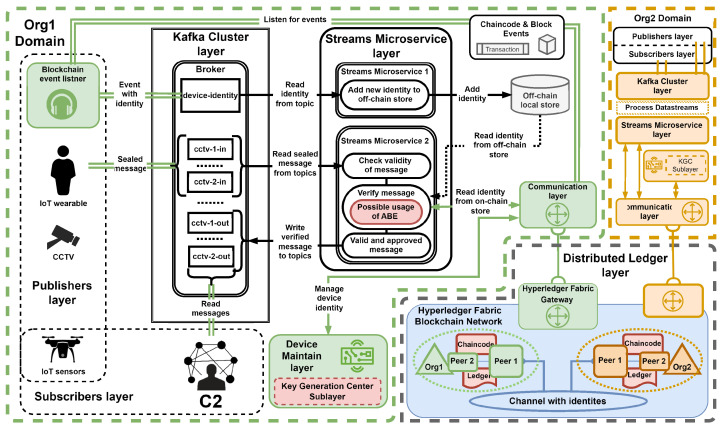
Detailed overview of the experimental environment.

**Figure 5 sensors-24-05863-f005:**
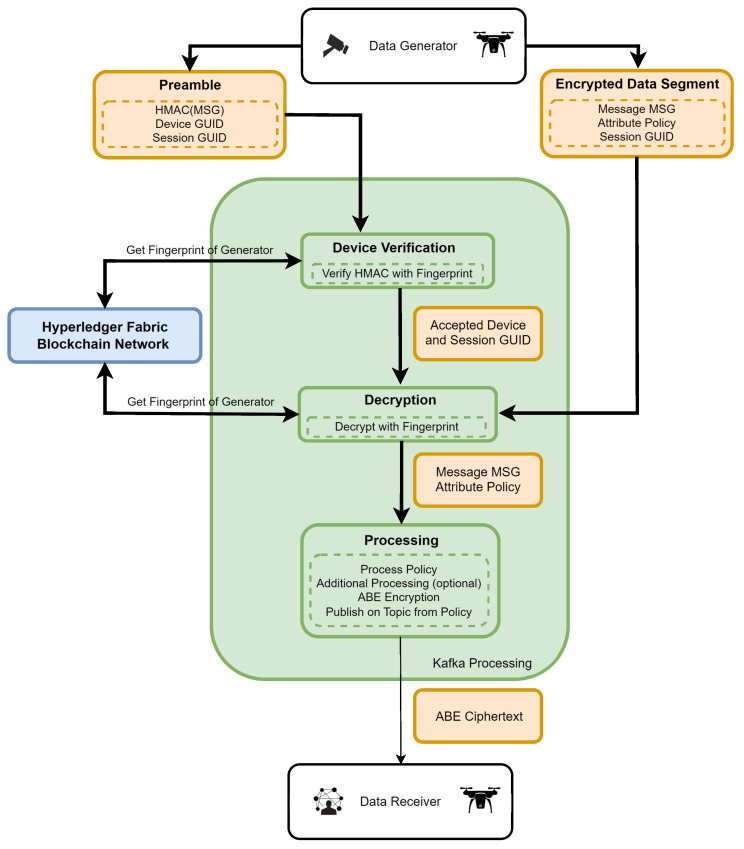
Diagram of data flow in proposed system.

**Figure 6 sensors-24-05863-f006:**
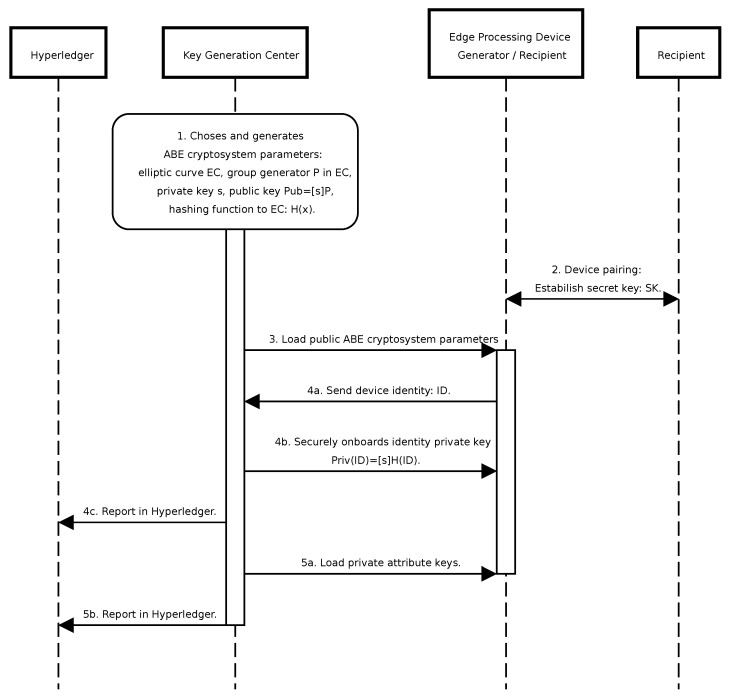
Sequence diagram illustrating the ABE system setup steps.

**Figure 7 sensors-24-05863-f007:**
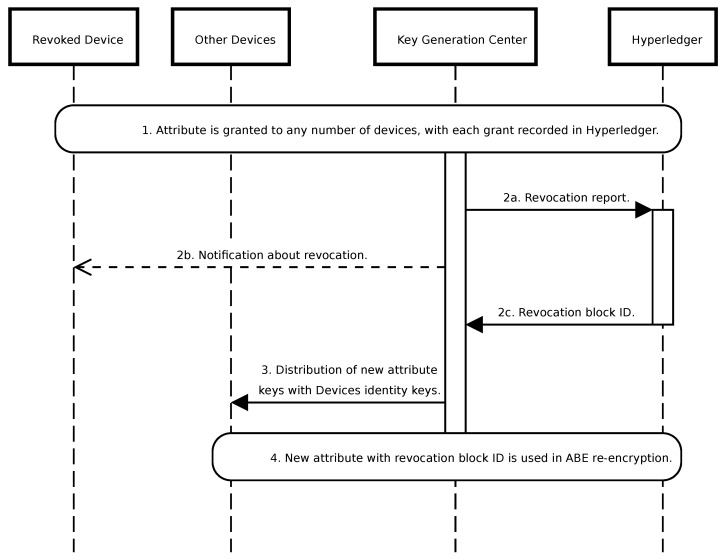
Sequence diagram illustrating the ABE attribute revocation steps.

**Figure 8 sensors-24-05863-f008:**
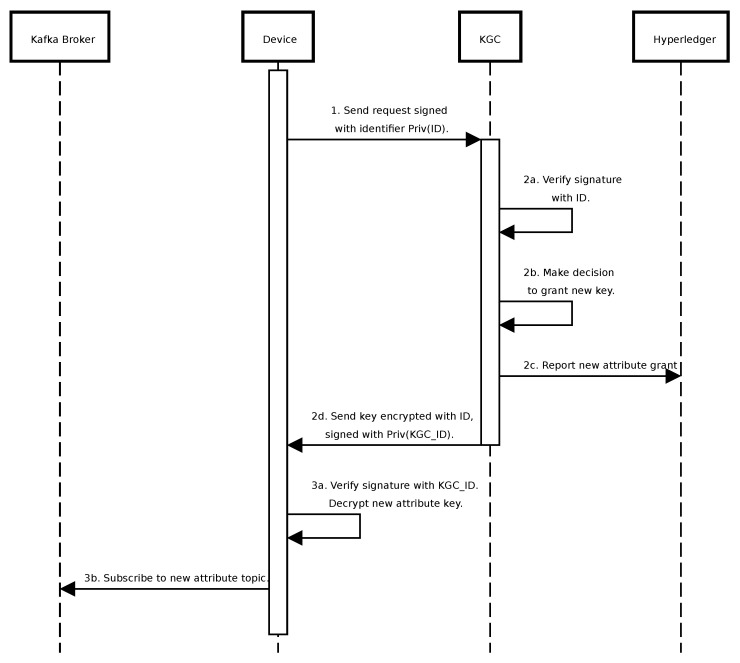
Sequence diagram of requesting new permissions.

**Figure 9 sensors-24-05863-f009:**
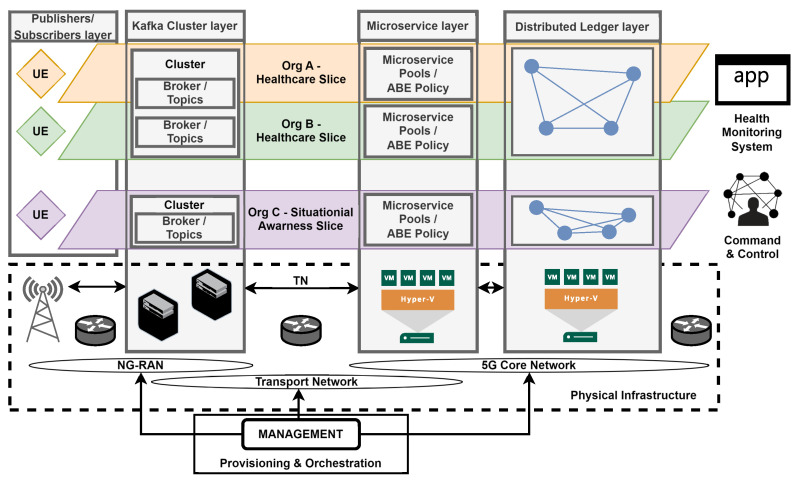
Deployment of data exchange system within 5G.

**Figure 10 sensors-24-05863-f010:**
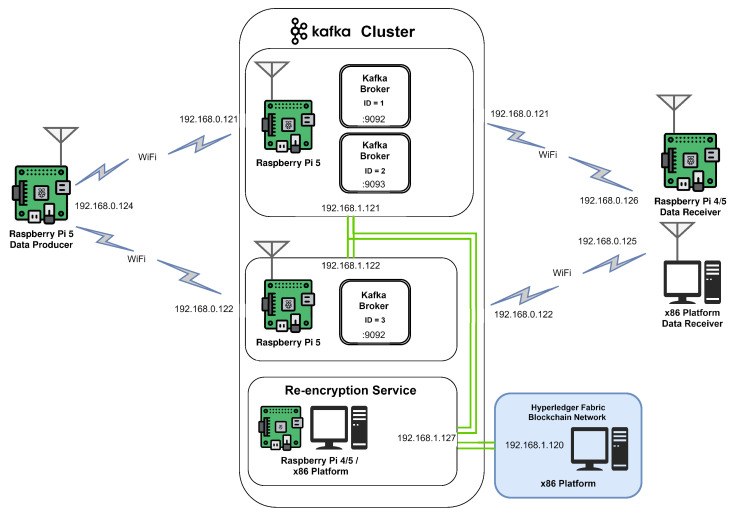
Scheme of experimental setup.

**Table 1 sensors-24-05863-t001:** Simulation results on x86 processor.

		Abe Encrypt				Abe Decrypt			
Message Length [bytes]	Number of Attributes	Percentile 0.9	Percentile 0.95	Percentile 0.99	Std Dev	Percentile 0.9	Percentile 0.95	Percentile 0.99	Std Dev
32	1	35.88 ms	35.85 ms	35.70 ms	0.5 ms	15.27 ms	15.35 ms	15.42 ms	0.3 ms
32	2	43.50 ms	43.40 ms	43.80 ms	0.5 ms	18.49 ms	18.12 ms	17.87 ms	0.3 ms
32	3	46.78 ms	46.64 ms	46.27 ms	0.7 ms	15.49 ms	15.57 ms	15.63 ms	0.3 ms
32	5	73.70 ms	73.34 ms	72.91 ms	0.8 ms	20.42 ms	20.20 ms	20.26 ms	0.4 ms
32	8	83.43 ms	83.72 ms	83.20 ms	0.9 ms	21.25 ms	21.29 ms	21.38 ms	0.4 ms

**Table 2 sensors-24-05863-t002:** Ciphertext length results depending on the number of attributes.

Message Length [bytes]	Number of Attributes	Ciphertext Length [bytes]
32	1	2384
32	2	2744
32	3	3093
32	5	4567
32	8	4858

**Table 3 sensors-24-05863-t003:** Simulation results on Raspberry Pi 4.

		Abe Encrypt				Abe Decrypt			
Message Length [bytes]	Number of Attributes	Percentile 0.9	Percentile 0.95	Percentile 0.99	Std Dev	Percentile 0.9	Percentile 0.95	Percentile 0.99	Std Dev
32	1	736.44 ms	736.41 ms	736.34 ms	14.1 ms	281.45 ms	281.47 ms	281.58 ms	7.2 ms
32	2	905.95 ms	906.01 ms	906.05 ms	14.7 ms	291.47 ms	291.42 ms	291.37 ms	7.7 ms
32	3	1.0780 s	1.0791 s	1.0786 s	15.2 ms	301.55 ms	301.57 ms	301.52 ms	8.1 ms
32	5	1.4257 s	1.4246 s	1.4251 s	16.3 ms	321.41 ms	321.49 ms	321.44 ms	8.3 ms
32	8	1.9390 s	1.9387 s	1.9393 s	26.2 ms	351.69 ms	351.75 ms	351.73 ms	8.9 ms

**Table 4 sensors-24-05863-t004:** Simulation results on Raspberry Pi 5.

		Abe Encrypt				Abe Decrypt			
Message Length [bytes]	Number of Attributes	Percentile 0.9	Percentile 0.95	Percentile 0.99	Std Dev	Percentile 0.9	Percentile 0.95	Percentile 0.99	Std Dev
32	1	444.75 ms	443.71 ms	443.14 ms	11.2 ms	201.76 ms	201.76 ms	201.74 ms	6.4 ms
32	2	550.68 ms	550.11 ms	549.76 ms	11.6 ms	213.71 ms	213.74 ms	213.73 ms	6.5 ms
32	3	657.90 ms	657.52 ms	656.59 ms	13.1 ms	226.44 ms	226.43 ms	226.42 ms	7.2 ms
32	5	862.46 s	861.99 ms	862.10 ms	14.7 ms	250.35 ms	250.35 ms	250.34 ms	7.3 ms
32	8	1.1826 s	1.1817 s	1.1814 s	17.2 ms	287.00 ms	287.00 ms	286.99 ms	8.0 ms

**Table 5 sensors-24-05863-t005:** Combined times of Kafka processing and ABE re-encryption. (a)—Intel i5-13600KF; (b)—Raspberry Pi 4.

Number of Attributes	Kafka and HL Processing—Avg Time	ABE Encrypt— Avg Time	Kafka and ABE Encrypt—Avg Time	ABE Decrypt— Avg Time		Kafka and ABE Encrypt and Decrypt— Avg Time	
		(a)		(a)	(b)	(a)	(b)
1	48.6 ms	35.70 ms	84.30 ms	15.35 ms	281.58 ms	99.65 ms	365.88 ms
2	48.6 ms	43.80 ms	92.4 ms	18.12 ms	291.37 ms	110.52 ms	383.77 ms
3	48.6 ms	46.27 ms	94.87 ms	15.57 ms	301.52 ms	110.44 ms	396.39 ms
5	48.6 ms	72.91 ms	121.51 ms	20.20 ms	321.44 ms	141.71 ms	442.95 ms
8	48.6 ms	83.20 ms	131.8 ms	21.29 ms	351.73 ms	153.09 ms	483.53 ms

**Table 6 sensors-24-05863-t006:** Combined times of Kafka processing and ABE re-encryption. (a)—Intel i5-13600KF; (b)—Raspberry Pi 5.

Number of Attributes	Kafka and HL Processing—Avg Time	ABE Encrypt— Avg Time	Kafka and ABE Encrypt—Avg Time	ABE Decrypt— Avg Time		Kafka and ABE Encrypt and Decrypt— Avg Time	
		(a)		(a)	(b)	(a)	(b)
1	48.6 ms	35.70 ms	84.30 ms	15.35 ms	201.89 ms	99.65 ms	286.19 ms
2	48.6 ms	43.80 ms	92.4 ms	18.12 ms	213.84 ms	110.52 ms	306.24 ms
3	48.6 ms	46.27 ms	94.87 ms	15.57 ms	226.58 ms	110.44 ms	321.45 ms
5	48.6 ms	72.91 ms	121.51 ms	20.20 ms	250.48 ms	141.71 ms	371.99 ms
8	48.6 ms	83.20 ms	131.8 ms	21.29 ms	287.15 ms	153.09 ms	418.95 ms

**Table 7 sensors-24-05863-t007:** Time for generating new private key depending on the number of attributes.

Number of Attributes	Intel i5-13600KF	Raspberry Pi 5
5	0.422 s	1.914 s
10	0.795 s	2.895 s
15	1.371 s	3.904 s
20	1.928 s	4.884 s

**Table 8 sensors-24-05863-t008:** Benchmark comparison between TKN20 and RSA results.

Algorithm	KeyGen		Encrypt		Decrypt		Enc + Dec	
	Mean	St. Dev.	Mean	St. Dev.	Mean	St. Dev.	Mean	St. Dev.
RSA 2048	84 ms	46 ms	0.018 ms	0.005 ms	0.5 ms	0.06 ms	0.5 ms	0.7 ms
RSA 3072	298 ms	145 ms	0.096 ms	0.021 ms	1.5 ms	0.08 ms	1.6 ms	0.1 ms
RSA 4096	927 ms	317 ms	0.164 ms	0.030 ms	3.5 ms	0.3 ms	3.7 ms	0.3 ms
TKN20 Short	28 ms	1 ms	26 ms	0.5 ms	9.7 ms	0.6 ms	36 ms	1.1 ms
TKN20 Long	72 ms	2 ms	41 ms	0.5 ms	7.5 ms	0.3 ms	49 ms	0.8 ms

**Table 9 sensors-24-05863-t009:** Ciphertext overhead length of 32 bytes encryption—comparison between TKN20 and RSA.

Algorithm	Ciphertext Size
RSA3072	489
TKN20 Short	4105
TKN20 Long	4779

## Data Availability

Data are contained within the article.

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
