# Peer review of "Application of Attribute-Based Encryption in Military Internet of Things Environment"

_sensors, 2024, doi:10.3390/s24185863_

Round 1
Reviewer 1 Report
Comments and Suggestions for Authors
This paper proposes a comprehensive solution for secure data dissemination by integrating Hyperledger Fabric’s distributed registry technology, Apache Kafka message broker, and data processing microservices implemented using the Kafka Streams API library. Additionally, attribute-based encryption (ABE), a cryptographic approach that facilitates controlling access to information based on data and user attributes rather than traditional methods that rely on specific keys, is applied. Detailed review suggestions are listed as follows:
The authors should highlight the novelty of their work. Why was ABE selected as the design approach in this scenario, and are there any alternative methods that can achieve the same goal more efficiently?
The experimental data is insufficient and the evaluation indicators are limited. Can you consider the impact of the system performance on time complexity and space complexity from multiple dimensions?
It is advisable to consult relevant literature in the field without a comparative scheme for comparative analysis.
Can the symbols used in the text be listed separately in a table to increase readability and facilitate readers' understanding?
Comments on the Quality of English Language
There are a few typographical errors that need correction, such as:
Page 16: The excessive white space before and after Figure 8 should be eliminated for improved presentation.
Author Response
We thank the Reviewer for their comments and professional advice. These comments are valuable and helpful for revising and improving our paper and the important guiding significance to our research. We have carefully studied the comments and have made corrections, which we hope will be met with approval. The revised sections are marked in yellow on the paper attached. The main corrections in the paper and the responses to the reviewer's comments are as following:
1. The authors should highlight the novelty of their work. Why was ABE selected as the design approach in this scenario, and are there any alternative methods that can achieve the same goal more efficiently?
The authors’ answer: We have improved the Introduction section. We have modified the description of requirements for a data distribution system in a federated MIoT environment included in the Introduction section. One of the basic requirements for MIoT systems is to meet the data-centric security paradigm, i.e., to ensure that data is securely managed from the point of origin to the end user. Meeting this requirement is possible with the use of an appropriate ABE scheme, which replaces, in traditional terms, the confidentiality of transmitted data and the authorization of operations performed on it. The use of the selected ABE scheme provides the possibility of fine-grained access control depending on the attributes of the IoT device, the content of the data itself, and the characteristics of the recipient and can be defined at the level of the federation's security policy. We also pointed out the novelty of the work in the Introduction part when describing our contribution.
2. The experimental data is insufficient and the evaluation indicators are limited. Can you consider the impact of the system performance on time complexity and space complexity from multiple dimensions?
The authors’ answer: We have added additional the experiment (Std Deviation) and conducted a new test with the statistics from the experiment (Std Deviation) and conducted a new test with the tic from experiment (Std Deviation) and conducted a new test with benchmarking algorithm we had used (TKN20) and RSA for comparison.
3. It is advisable to consult relevant literature in the field without a comparative scheme for comparative analysis.
The authors’ answer: We have cited literature to compare with ABAC systems in general. We have conducted new benchmarking tests to compare algorithm we had used (TKN20) and RSA used in comparative schemes as part of PKI.
4. Can the symbols used in the text be listed separately in a table to increase readability and facilitate readers' understanding?
The authors’ answer: In response we have added Abbreviations subsection at the end of article for explaining the abbreviations used.
5. There are a few typographical errors that need correction, such as Page 16: The excessive white space before and after Figure 8 should be eliminated for improved presentation.
The authors’ answer: Changed the image size to eliminate white space.
We tried our best to improve the manuscript and made some changes in the manuscript. These changes will not influence the content and framework of the paper.
We appreciate Editors/Reviewers’ earnest work and hope that the correction will meet with approval.
Once again, thank you very much for your comments and suggestions.
Sincerely,
Authors

Reviewer 2 Report
Comments and Suggestions for Authors
The paper still needs improvement in this journal.
1. The use of many abbreviations without clarifying what they mean.
2. The entire manuscript should be further polished. The organization and logic flow of this paper should be refined.
3. In the abstract and introduction, authors list many issues and challenges, but this paper does not highlight which challenges and issues this paper is targeting for. And the motivation of this paper is too high level, and need to narrow down to a concrete research problem.
4. The introduction part should be re-organized and properly segmented to make it more layered. The motivation of the work should be further underlined.
5-The proposed work should be compared with the relevant studies.
6-Authors need to review the D2D for 5G based on VANET system as well by citing the following articles. -Lattice-Based Lightweight Quantum Resistant Scheme in 5G-Enabled Vehicular Networks; -FC-PA: Fog Computing-based Pseudonym Authentication Scheme in 5G-enabled Vehicular Networks;
7 What is the main difference between the proposal and others?
8-How this solution outperforms others.
9-The proposal solution should be explained by adding more figures/steps
10-Provide insights into potential future directions for research or practical applications based on the study's outcomes.
11-Ensure technical terms and concepts are well-defined or explained for readers who may not be experts in the field.
Comments on the Quality of English Language
need improve
Author Response
We thank the Reviewer for their comments and professional advice. These comments are valuable and helpful for revising and improving our paper and the important guiding significance to our research. We have carefully studied the comments and have made corrections, which we hope will be met with approval. The revised sections are marked in yellow on the paper attached. The main corrections in the paper and the responses to the reviewer's comments are as following:
1. The use of many abbreviations without clarifying what they mean.
The authors’ answer: In response we have added Abbreviations subsection at the end of article for explaining the abbreviations used.
2. The entire manuscript should be further polished. The organization and logic flow of this paper should be refined.
The authors’ answer: We have thoroughly reviewed the article and corrected the parts concerning the rationale and motivation for the work, the extent to which the subject matter is related to other similar work, and comparing the performance results obtained with those for a PKI-based solution.
3. In the abstract and introduction, authors list many issues and challenges, but this paper does not highlight which challenges and issues this paper is targeting for. And the motivation of this paper is too high level, and need to narrow down to a concrete research problem.
The authors’ answer: We introduced our solution to the entire framework for secure data distribution in a federated MIoT environment. In response to this comment, we indicated what we focused on in the article in the modified Introduction section and improved the abstract in this context.
4. The introduction part should be re-organized and properly segmented to make it more layered. The motivation of the work should be further underlined.
The authors’ answer: In response we reorganized and changed some parts of the Introduction section.
5. The proposed work should be compared with the relevant studies.
The authors’ answer: We have cited literature to compare with ABAC systems in general. We have conducted new benchmarking tests to compare algorithm we had used (TKN20) and RSA used in comparative schemes as part of PKI.
6. Authors need to review the D2D for 5G based on VANET system as well by citing the following articles. -Lattice-Based Lightweight Quantum Resistant Scheme in 5G-Enabled Vehicular Networks; -FC-PA: Fog Computing-based Pseudonym Authentication Scheme in 5G-enabled Vehicular Networks.
The authors’ answer: In response we have looked into these D2D for 5G solutions and articles, found them very interesting and decided to cite them in related works section.
7. What is the main difference between the proposal and others?
The authors’ answer: In response we have created section PKI Comparison comparing our solution to existing ABAC systems.
8. How this solution outperforms others.
The authors’ answer: we have conducted new benchmarking tests to compare algorithm we had used (TKN20) and RSA used in comparative schemes as part of PKI and shown under what circumstances it becomes beneficial to use ABE.
9. The proposal solution should be explained by adding more figures/steps
The authors’ answer: In response we believe that solution is detailed with text and sequence diagrams in every solution process: Setup, Key Generation, Key Distribution, Key Revocation, Encryption and Decryption; however we have added additional sections further explaining nuances regarding the system.
10. Provide insights into potential future directions for research or practical applications based on the study's outcomes.
The authors’ answer: In response we have added new sections further explaining practical applications within MIoT and 5G networks.
11. Ensure technical terms and concepts are well-defined or explained for readers who may not be experts in the field.
The authors’ answer: In response we have added more explanations and modified existing ones as well as added an Abbreviations section at the end of an article.
We tried our best to improve the manuscript and made some changes in the manuscript. These changes will not influence the content and framework of the paper.
We appreciate Editors/Reviewers’ earnest work and hope that the correction will meet with approval.
Once again, thank you very much for your comments and suggestions.
Sincerely,
Authors

Reviewer 3 Report
Comments and Suggestions for Authors
In my opinion, the paper seems important regarding Security issues in the Military Internet of Things. The authors present new solutions for secure data dissemination. The paper is well-written and makes a proper scientific contribution to the area considered. I suggest to publish the paper.
Author Response
We thank the Reviewer for their comments and professional advice. These comments are valuable and helpful for revising and improving our paper and the important guiding significance to our research.
"In my opinion, the paper seems important regarding Security issues in the Military Internet of Things. The authors present new solutions for secure data dissemination. The paper is well-written and makes a proper scientific contribution to the area considered. I suggest to publish the paper."
The authors’ answer: Many thanks for recognizing our efforts to improve MIoT network security.
Once again, thank you very much for your comments and suggestions.
Sincerely,
Authors
Reviewer 4 Report
Comments and Suggestions for Authors
The authors presented a new mobile network security scheme. The research aims to demonstrate the usefulness of such a scheme for protecting military networks. The presented results are new and exciting. However, before publication, the authors should consider two aspects of modern mobile networks. First, the presented military network is a system of several different networks, as the authors themselves present in the introductory part of the work (V2X, e-health, etc.). From a business point of view, these networks are (5G) virtual industries with their management requirements, especially access management. Second, the possibility of implementing verticals with the required quality of service requires the use of network slicing with appropriate security (isolation). The publication would show how the security scheme will work in several vertical industries implemented in parallel network slices.
Author Response
We thank the Reviewer for their comments and professional advice. These comments are valuable and helpful for revising and improving our paper and the important guiding significance to our research. We have carefully studied the comments and have made corrections, which we hope will be met with approval. The revised sections are marked in yellow on the paper. The main corrections in the paper and the responses to the reviewer's comments are as following:
1. First, the presented military network is a system of several different networks, as the authors themselves present in the introductory part of the work (V2X, e-health, etc.). From a business point of view, these networks are (5G) virtual industries with their management requirements, especially access management.
The authors’ answer: In response we have added a new section presenting a general proposal for how our data exchange system (security scheme) will work in several vertical industries implemented in parallel network slices.
2. Second, the possibility of implementing verticals with the required quality of service requires the use of network slicing with appropriate security (isolation). The publication would show how the security scheme will work in several vertical industries implemented in parallel network slices.
The authors’ answer: In response we have added a new section presenting possible infrastructure deployments of our data exchange system within 5G networks considering network slicing and isolation mechanisms for specific vertical industries.
We tried our best to improve the manuscript and made some changes in the manuscript. These changes will not influence the content and framework of the paper.
We appreciate Editors/Reviewers’ earnest work and hope that the correction will meet with approval.
Once again, thank you very much for your comments and suggestions.
Sincerely,
Authors

Round 2
Reviewer 2 Report
Comments and Suggestions for Authors
No additional comments
Comments on the Quality of English Languageno